# Botanical Collection Patterns and Conservation Categories of the Most Traded Timber Species from the Ecuadorian Amazon: The Role of Protected Areas

**DOI:** 10.3390/plants12183327

**Published:** 2023-09-20

**Authors:** Rolando López-Tobar, Robinson J. Herrera-Feijoo, Rubén G. Mateo, Fernando García-Robredo, Bolier Torres

**Affiliations:** 1Facultad de Ciencias Agrarias y Forestales, Universidad Técnica Estatal de Quevedo (UTEQ), Quevedo Av. Quito km, 1 1/2 Vía a Santo Domingo de los Tsáchilas, Quevedo 120550, Ecuador; rlopez@uteq.edu.ec; 2Escuela Técnica Superior de Ingeniería de Montes, Forestal y del Medio Natural, Universidad Politécnica de Madrid, 28040 Madrid, Spain; 3Unidad de Posgrado, Universidad Técnica Estatal de Quevedo (UTEQ), Quevedo Av. Quito km, 1 1/2 Vía a Santo Domingo de los Tsáchilas, Quevedo 120550, Ecuador; 4Escuela de Doctorado, Centro de Estudios de Posgrado, Universidad Autónoma de Madrid, C/Francisco Tomás y Valiente, nº 2, 28049 Madrid, Spain; 5Departamento de Biología (Botánica), Facultad de Ciencias, Universidad Autónoma de Madrid, 28049 Madrid, Spain; rubeng.mateo@uam.es; 6Centro de Investigación en Biodiversidad y Cambio Global (CIBC-UAM), Facultad de Ciencias, Universidad Autónoma de Madrid, 28049 Madrid, Spain; 7Departamento de Ingeniería y Gestión Forestal y Ambiental, Escuela Técnica Superior de Ingeniería de Montes, Forestal y del Medio Natural, Universidad Politécnica de Madrid, C/José Antonio Novais 10, 28040 Madrid, Spain; fernando.garcia.robredo@upm.es; 8Facultad de Ciencia de la Vida, Universidad Estatal Amazónica (UEA), Puyo 160101, Ecuador; btorres@uea.edu.ec; 9Ochroma Consulting and Services, Puerto Napo, Tena 150150, Ecuador

**Keywords:** collection patterns, botanical collections, SNAP, forest extraction, IUCN

## Abstract

The Ecuadorian Amazon is home to a rich biodiversity of woody plant species. Nonetheless, their conservation remains difficult, as some areas remain poorly explored and lack georeferenced records. Therefore, the current study aims predominantly to analyze the collection patterns of timber species in the Amazon lowlands of Ecuador and to evaluate the conservation coverage of these species in protected areas. Furthermore, we try to determine the conservation category of the species according to the criteria of the IUCN Red List. We identified that one third of the timber species in the study area was concentrated in three provinces due to historical botanical expeditions. However, a worrying 22.0% of the species had less than five records of presence, and 29.9% had less than ten records, indicating a possible underestimation of their presence. In addition, almost half of the species evaluated were unprotected, exposing them to deforestation risks and threats. To improve knowledge and conservation of forest biodiversity in the Ecuadorian Amazon, it is recommended to perform new botanical samplings in little-explored areas and digitize data in national herbaria. It is critical to implement automated assessments of the conservation status of species with insufficient data. In addition, it is suggested to use species distribution models to identify optimal areas for forest restoration initiatives. Effective communication of results and collaboration between scientists, governments, and local communities are key to the protection and sustainable management of forest biodiversity in the Amazon region.

## 1. Introduction

Ecuador is considered one of the most biodiverse countries on the planet [1,2]. Two of the five most important biodiversity hotspots for biological conservation in South America are located in this country, being the tropical Andes and the Tumbes-Chocó-Magdalena Corridor [3,4]. The country boasts a remarkable diversity of endemic vascular plant species (approximately 26%; [5]) as well as being a sanctuary for various mammals [6], amphibians [7], reptiles [8], and birds [9]. Moreover, Ecuador’s varied topography encompasses a wide array of ecosystems, including the Páramo (alpine tundra), mangroves, cloud forest, and tropical jungle. These distinct habitat types give rise to three main biomes: the Coastal lowlands, the Andean highlands, and the Amazon basin. However, despite its ecological significance, Ecuador has faced an alarming rate of deforestation. According to the historical deforestation map of the country, the loss of forest cover has been accelerating over the years. The records indicate it had 14,587,770.6 ha (58.59%) in the year 1990, 13,660,353.63 ha (54.87%) in 2000, 13,038,367.32 ha (52.37%) in 2008, 12,753,386.91 ha (51.22%) in 2014, 12,631,197.69 ha (50.73%) in 2016, and 12,514,339.53 (50.26%) in 2018. The most recent evaluation in 2020 paints a disheartening picture, with only 48.97% of native forest remaining [10]. This worrisome trend in deforestation demands urgent attention and conservation efforts. The significant loss of natural habitats can have profound implications for the biodiversity of Ecuador. Addressing this issue and implementing sustainable practices is crucial to preserving Ecuador’s unique ecosystems and safeguarding its standing as a biodiversity hotspot.

Within this epicenter of biodiversity lies the Ecuadorian Amazon Region (EAR), hailed as one of the most biodiverse areas on the planet [11,12,13], especially for its high diversity of woody plants [14,15,16,17,18,19]. This biologically rich region is politically divided into six provinces: Sucumbíos, Orellana, Napo, Pastaza, Morona Santiago, and Zamora Chinchipe. These provinces play a fundamental role in providing wood supply, primarily for the national market. Interestingly, certain local populations have adopted livelihood strategies centered around the use of native forest wood, for whom the average income per household can reach up to 50% of the total income [20]. This includes migrant settlers and indigenous communities engaged in both formal and informal logging activities [21].

The forests and productive systems within the EAR also hold tremendous potential for its inhabitants, offering a myriad of opportunities linked to essential ecosystem services [20,22]. One such opportunity involves initiatives aimed at incentivizing carbon capture [23,24], along with existing programs like the Socio Bosque Program [25]. Specifically, the PSB is a conservation initiative based on the protection of Ecuador’s forests, which provides economic incentives to peasants and indigenous communities willing to protect and conserve their native forests, marshlands, or other native vegetation [26]. These incentives are granted on the basis of a prior agreement in which the landowners undertake to conserve the registered area for a period of 20 years [10]. In Ecuador, there are 2723 polygons under conservation within the PSB, belonging to 2335 landowners; these areas cover 6.7% of continental Ecuador. Specifically, the ARE has 1069 polygons under conservation (39% of the total number of polygons), covering an area of 1,374,026.9 hectares, which represents 82.3% of the national PSB and 11.8% of the total area of the Amazon region [10].

However, these forests have been suffering strong deforestation processes caused mainly by the change in land use [27,28,29] and the conversion of land for agricultural purposes [30,31,32]. In this regard, Ecuador is one of the Latin American countries with the highest reported deforestation rate in the period 1990–2010, with an average annual loss of 1.5% and −1.8% of native forest cover [33,34]. A recent study suggests that approximately 454,000 hectares of native forest were deforested in the 64 ecosystems between 1990 and 2014 [31]. Specifically in relation to the AER, it was found that this was the region of Ecuador with the highest gross deforestation rate in the period 2016–2018 [35].

These activities can be traced back to the Agrarian Reform and Colonization Law in the 1960s and 1970s [36], which marked the beginning of such transformations, further exacerbated by the formal and informal extraction of wood [37].

The ongoing deforestation and the scarcity of comprehensive botanical records pose significant challenges in the identification, monitoring, and effective conservation of forest species within the EAR. A study conducted by Guevara et al. [38] suggests that about 71.8% of the tree species identified in the lowlands have fewer than five observations with no geographic duplicates. This lack of information is critical because conservation efforts become hindered without understanding a species’ geographic distribution patterns and biotic interactions within its habitat [39]. Therefore, it requires rapid responses to reverse the current rates of extinction, urging an increase in collection sampling and subsequent digitalization of data to make it freely accessible to the scientific community [40]. To address this issue, the georeferenced data available in herbaria provide invaluable information to clarify the distribution patterns of tree species and assess their current risk of extinction [41,42]. By consolidating and sharing such data, researchers can gain crucial insights into the geographical prevalence of species and devise effective conservation strategies. This collective effort will be instrumental in fostering a deeper understanding of the region’s biodiversity and aiding the preservation of its invaluable forest ecosystems.

In Ecuador, there are various in situ conservation initiatives, emphasizing the importance of understanding timber species collection patterns and their relationships with conservation for effective natural resource management in the EAR. These collection patterns provide valuable information into the geographic distribution and relative abundance of species of commercial interest, which, in turn, allows one to assess the effectiveness of current in situ conservation measures implemented in a megadiverse country like Ecuador, such as the National System of Protected Areas (SNAP, for its Spanish acronym), the Protective Forests and Vegetation (BVP, for its Spanish acronym), the Socio Bosque Program (PSB, for its Spanish acronym), and the State Forest Patrimony (PFE, for its Spanish acronym). A recent study revealed that 72% (4437 species) of threatened endemic vascular plants in Ecuador exist outside the protection boundaries of the SNAP [43]. While the SNAP remains the most effective in situ conservation strategy to preserve biodiversity in Ecuador [44,45], it becomes crucial to explore the relationship between these conservation categories and the collection patterns of timber-yielding forest species of high commercial interest.

Currently, the risk of extinction has only been assessed for 10% of plant species globally, equivalent to 61,371 species [46,47]. In the context of Ecuador, 5460 plant species have been assessed, of which 2711 are endemic. Of this group, 2766 species have been assessed as being in the category of least concern, 992 are considered vulnerable, 769 are endangered, and 275 are critically endangered [47]. On the other hand, a previous study by Guevara et al. [38], with a database of 47,486 individuals collected from lowland trees in the EAR, identified 106 species as currently classified as Least Concern, 78 as Vulnerable, 16 as Endangered, 9 as Data Deficient, and 1 as Critically Endangered. On the other hand, the same study found that 89% of lowland tree species in the EAR are currently listed as Not Evaluated. Given that only a fraction of plant species have been assessed for global extinction risk, and given the richness of biodiversity in the Ecuadorian Amazon, it is essential to have up-to-date and detailed knowledge of the conservation status of the most traded timber species in this region. This knowledge is not only essential for making informed conservation and natural resource management decisions but can also provide critical insight into the potential impacts of human activities on these species and their habitats.

In this context, the present study aims to: (1) To characterize the botanical collection patterns of the most commercialized timber species in the Ecuadorian Amazon; (2) identify the current level of protection for the species analyzed within each of the conservation initiatives in force in Ecuador; and (3) examine the current IUCN conservation category for all species used in this study.

## 2. Results

### 2.1. Collection Patterns of Harvested Timber Species from the Ecuadorian Amazon

In total, the database analyzed consisted of 12,992 occurrence records for 214 taxa, providing a solid basis for analyzing the geographical patterns of harvesting of the selected timber species in the EAR. Among these records, the ten species with the highest number of observations were *Guarea kunthiana* (481), *Grias neuberthii* (329), *Mayna odorata* (322), *Guarea macrophylla* (296), *Tapirira guianensis* (287), *Dacryodes peruviana* (220), *Matisia malacocalyx* (206), *Minquartia guianensis* (192), *Alchornea glandulosa* (175), and *Apeiba membranacea* (174) (See Appendix A). However, it is worth noting that the analysis also revealed that 47 species (22.0%) had less than five georeferenced records of presence, indicating limited data availability for these species. Additionally, for 64 species (29.9%), less than ten georeferenced records were found, further highlighting the relatively sparse presence information for a substantial portion of the analyzed timber taxa. These findings underscored the varying degrees of data availability for the timber species, with some exhibiting a robust number of observations, whereas others suffered from limited representation in the georeferenced records.

Regarding the spatial distribution of the records collected for the 214 taxa analyzed, the results showed that 33% (4286) of the individuals reported for the Ecuadorian Amazon were found in the province of Orellana, followed by the province of Sucumbíos with 1864 (14.3%), and Napo with 1800 (13.9%) (Table 1 and Figure 1). However, upon closer examination at the species level, our analysis unveiled interesting patterns. The Province of Morona Santiago emerged as the area with the highest diversity of species, housing 199 distinct timber species. Sucumbíos and Orellana followed closely, with 146 species each, recorded per individual.

### 2.2. Protection Coverage of Timber Species within Current Conservation Initiatives

Figure 2 presents the results of our analysis concerning the current protection coverage of the timber species. The findings reveal that the highest level of protection is provided by the SNAP, which currently protects 24.5% (167 species) of the considered population. Additionally, we determined that the BVP offers coverage to 10.2% (151 species) of the individuals analyzed, whereas the properties assigned to the category of PFE contribute to the protection of 9% (120 species), followed by the SB with 9% (138 species). On the other hand, it was found that 48.3% (209 species) are currently outside the four established protection categories.

### 2.3. IUCN Conservation Categories for the Analyzed Timber Species

Figure 3 provides a comprehensive overview of the current IUCN conservation categories assigned to the 214 analyzed timber taxa within the EAR. In general terms, the results suggest that 142 species (66.4%) (See Appendix A) would be classified as Least Concern (LC), suggesting that they are not currently considered to be facing significant threats to their survival. However, the analysis also revealed that a small proportion of the species fell into more critical conservation categories. Seven species (3.3%) are classified as Vulnerable (VU), being *Cedrela odorata*, *Prumnopitys montana*, *Aniba perutilis*, *Humiriastrum procerum*, *Guarea cartaguenya*, *Jacaranda mimosifolia*, and *Nectandra guadaripo*. Furthermore, two species (0.9%) were identified as Endangered (EN), comprising *Cedrela odorata* and *Cedrela fisilis*. Additionally, one species (0.5%) was reported as Critically Endangered (CR), being *Centrolobium ochroxylum*. In addition, two species (0.9%) classified as Data Deficient (DD) were found, being *Pachira rupicola* and *Pseudobombax millei*. These species require further research and data collection to assess their conservation status accurately. Finally, it is important to note that a substantial portion of the evaluated species, sixty in total (28.0%), are currently not considered in the IUCN database. As a result, they do not fall into any of the current conservation categories, highlighting the need for further research and assessment to determine their conservation status.

Figure 4 depicts the distribution of the analyzed species across the protection categories in both Ecuador and the (IUCN) classifications. The results reveal that species falling into all IUCN conservation categories receive protection within the National System of Protected Areas (SNAP) in Ecuador. Specifically, 127 species classified as Least Concern (LC), 36 as Near Threatened (NE), 3 as Vulnerable (VU), and 1 as Endangered (EN) are encompassed within the SNAP. However, a significant number of species remain unprotected and are not covered by any of the conservation initiatives in force in Ecuador. This includes 141 species classified as Least Concern (LC), 57 as Near Threatened (NE), and 7 as Vulnerable (VU), being *Cedrela odorata*, *Prumnopitys montana*, *Aniba perutilis*, *Humiriastrum procerum*, *Guarea cartaguenya*, *Jacaranda mimosifolia*, and *Nectandra guadaripo*. There are also two endangered species (EN), *Matisia coloradorum* and *Swartzia littlei*. In addition, there is one Data Deficient (DD) species, which is *Pseudobombax millei*, whereas one is a Critically Endangered (CR) species, the *Centrolobium ochroxylum*.

## 3. Discussion

### 3.1. Collection Patterns across Provinces

The EAR is renowned for its exceptional biodiversity, particularly in terms of woody plant species [2,38,48]. Extensive botanical research has been conducted in this region, leading to a comprehensive understanding of arboreal communities and making it one of the most well-studied areas for vascular plants in the Amazon [18,49,50]. However, geographical knowledge of species distribution in tropical regions often remains limited or absent due to sampling biases [51,52,53,54]. These biases may be particularly associated with collections at sites that are biologically attractive to the collector due to their importance for global conservation [55] or locations that are easily accessible, such as those close to rivers and roads. Regarding the collection patterns of timber species in the six provinces of the EAR, our analysis revealed a notable concentration of records in certain areas. Specifically, a significant portion of the collected individuals was found in the Provinces of Orellana (33%), Sucumbíos (14.3%), and Napo (13.9%). These patterns can be attributed, in part, to historical botanical expeditions of Swedish, Danish, and American researchers in the 1970s and 1990s, which primarily focused on these provinces [56]. Consequently, these areas received more attention, resulting in a higher number of collected records for timber species. However, it is important to consider that these collection patterns may not fully represent the true distribution of timber species across the EAR. Potential biases introduced by historical expeditions and accessibility constraints may have influenced the spatial representation of the collected records. To gain a more comprehensive understanding of the distribution of timber species in the region, further research and data collection efforts are needed, aiming to reduce sampling biases and enhance geographical knowledge across all provinces of the Ecuadorian Amazon.

Regarding the species with the highest number of observations in our study, we observed similar results to those reported by Guevara et al. [38], who analyzed 47,486 specimens collected in the EAR. They also found that the most collected tree species were *Matisia malacocalyx* (0.56%), *Guarea kunthiana* (0.41%), and *Guarea macrophylla* (0.40%), which could have been due to accessibility, notoriety, or scientific interest. The higher frequency of collection for these species could be attributed to various factors, such as their accessibility, notoriety among collectors, or scientific interest due to specific characteristics or ecological significance.

In our study, we encountered a concerning lack of georeferenced records for a significant portion of the analyzed species. Specifically, we found that 22.0% of the species analyzed had less than five records of presence, whereas 29.9% had less than ten records. These figures were consistent with data reported globally by Enquist et al. [57], who estimated that approximately 36% of plants lacked adequate information on their distribution in global herbaria, and between 11.2% and 36.5% of these species had fewer than five reported observations. Similarly, a study focused on Ecuadorian plant species by Engemann et al. [58], using 205,735 specimens of 15,788 plant species, indicated a similar pattern, with half of the species having fewer than five observations. Additionally, when examining the tree presence database reported by Guevara et al. [38], we observed that about 71.8% of the trees in the list had fewer than five observations with no geographic duplicates. These results suggested that the presence of some species in the region could have been underestimated or unknown due to the paucity of available data. The limited availability of georeferenced records was fundamentally linked to accessibility issues. Sampling sites that were more accessible, such as those situated near roads or rivers, tended to be more frequently visited and sampled [59,60]. In contrast, remote or difficult-to-reach areas may remain underrepresented in botanical collections, leading to an incomplete understanding of the distribution of plant species in these regions.

### 3.2. Collection Patterns and Coverage Extent: The Role of Protected Areas

Currently, Ecuador maintains four conservation initiatives, SNAP, BVP, PFE, and SB, allowing protection coverage of 52.4% of the territory in the EAR [10]. Therefore, due to the importance of these initiatives for the conservation of biodiversity and forest resources, this study determined the conservation coverage of the most traded forest species within these initiatives.

Regarding the SNAP, our findings yielded that this initiative allowed the safeguarding of 24.5% of the analyzed species. These collection patterns within the SNAP have already been previously reported since a large part of the botanical collection efforts historically conducted in the EAR have been performed in the Yasuní National Park and Biosphere Reserve with the same name [11,61,62,63] mainly situated in the Province of Orellana. Meanwhile, with respect to the Province of Sucumbíos as reported by Guevara et al. [38], it is estimated that the Cuyabeno Faunistic Reserve located in this province is the second area that reports the highest number of botanical collections in the EAR [14,19]. Finally, in relation to the Province of Napo, the Sumaco Biosphere Reserve and six SNAP-protected areas (Cayambre Coca National Park, Sumaco Napo-Galeras National Park, Llanganates National Park, part of the Cotopaxi National Park, Antisana Ecological Reserve, and Colonso Chalupas Biological Reserve), where various investigations have been reported focused on characterizing the floristic biodiversity present in the area, are located in this region [13,49,64]. Nonetheless, in what corresponds to the number of species, it was evidenced that the Province of Morona Santiago harbors the largest number. This sampling effort can be related to the fact that there are two sites of great interest for conservation in this area, such as the Kutukú, the Shaimi Cordillera Protected Forest, and the Cordillera del Cóndor. Both sites have been fully studied for their biological wealth [65,66,67,68] and are considered priority areas for conservation [43].

Overall, our findings emphasize the pivotal role of protected areas in contributing to the collection patterns and conservation coverage of timber species in the EAR. These initiatives play a critical role in safeguarding biodiversity and promoting sustainable management practices within the region. Enhancing the protection and management of these areas will be essential for ensuring the long-term survival of valuable timber species and maintaining the ecological integrity of the Ecuadorian Amazon.

Meanwhile, regarding the BVP initiative, our results indicate that this initiative offers a coverage percentage of 10.2% to the species analyzed. The collection patterns identified demonstrated a greater collection effort within seven BVP. For example, in the province of Sucumbíos, the Pañacocha Protected Forest stands out [69], followed by the Cerro Sumaco Protected Forest located within the Orellana and Napo Provinces [70,71], whereas the Pablo López del Oglán and La Moravia Protected Forests stand out in the Pastaza Province [72,73]. On the other hand, for the province of Morona Santiago, a greater collection was observed within the Cordilleras Kutukú, Shaimi, and Tinajillas protective forests [74,75]. Finally, a greater number of botanical collections was evidenced in the Cuenca Alta del Río Nangaritza Protected Forest belonging to the Province of Zamora Chinchipe [76].

Regarding the PFE, our data revealed that this initiative contributed 9% of protection coverage at the EAR level. In this sense, it was possible to demonstrate that the collection patterns were registered in the Provinces of Sucumbíos, Orellana, and Napo. These patterns may be associated first with the high diversity reported in these areas and with the presence of global conservation and sustainable development strategies, such as the Yasuní Biosphere Reserves in Orellana, the Sumaco Biosphere Reserve in Sucumbíos, Napo, and part of Orellana, where multiple floristic inventories have historically been carried out [11,13,19,38,71].

Similarly, in what corresponds to the SB program, it was possible to determine that this initiative, like the PFE, would allow the analyzed timber species to have a 9% protection coverage within the administrative limits of the EAR. Therefore, it was possible to verify that the presence of properties included within the SB initiative in the six provinces of the EAR [10]. However, most of these properties and the collection effort are focused on the Province of Pastaza [73]. This may be associated with the fact that the province of Pastaza stands out for being the most extensive and still maintaining 94.1% of native forest (2,805,232.17%) [10], where 35% of the territory of the Yasuní National Park and 7 of the 14 indigenous nationalities of Ecuador are located [77,78]. This has resulted in the largest amount of territory in the PSB being found in this province, which is a positive example of payments for environmental services (PES) for forest conservation and where greater sampling efforts can be generated.

When analyzing the SNAP as a conservation initiative, the results indicate that this conservation category is currently likely to conserve 24.5% of the species studied, whereas the remaining 75.5% would be outside any conservation area. These conservation coverage values are similar to those reported by Cuesta et al. [43] in a study conducted for continental Ecuador, suggesting that 72% (4437 species) of Ecuador’s threatened endemic vascular plants are outside the boundaries of SNAP protection. This information is alarming, as unprotected species would be vulnerable to illegal logging, other anthropogenic threats, and even the effects of climate change [22,45,79]. However, when analyzing the coverage of the four conservation initiatives (SNAP, BVP, PSB, and PFE), our results show a reduction of up to 48.3% (209 species) of the species analyzed that are currently unprotected within these categories in Ecuador. In this context, the results obtained highlighted the importance of promoting and implementing other conservation options in addition to the SNAP for the conservation of forest resources, such as the PSB, BVP, and PFE.

This lack of protection provides a unique opportunity to promote the conservation of the analyzed timber species through cooperation with local communities [80,81]. In general, currently, trees outside forests (TOFs) are of great importance in developing regions, as they are often the main source of timber resources for local populations [82]. In this context, TOFs play a vital role in providing essential sources of timber, fuel, food, and income [83]. In addition, TOFs can play a crucial role as forest substitutes, providing rural communities with access to essential livelihood products such as fuelwood, fodder, and timber [84,85]. Harnessing this opportunity not only provides economic support to local communities, reducing their dependence on external purchasing power, but also promotes job creation and strengthens the resilience of these populations [85,86]. On the other hand, TOFs offer numerous opportunities focused on climate change mitigation due to the ecosystem services they provide, such as carbon storage [87].

To ensure the long-term conservation of unprotected timber species, it is essential to implement strategies that actively involve local communities in sustainable forest management. This could include training programs, promotion of sustainable agricultural and forestry practices, and raising awareness of the importance of biodiversity conservation. In addition, it is important to develop policies and regulations that support these initiatives and encourage collaboration between government sectors, NGOs, and local communities to achieve a holistic approach to conservation that benefits both biodiversity and human well-being.

### 3.3. IUCN Conservation Categories

The IUCN Red List criteria provide valuable insights into the conservation statuses of species worldwide. Global estimates suggest that between 36% and 57% of Amazonian trees may be threatened according to this criteria [88]. In our study focusing on the EAR, we found that 66.4% (142 species) of examined timber species fell into the “Least Concern” (LC) category, 3.3% were classified as “Vulnerable” (VU), and two species (0.9%) were listed as “Endangered” (EN). These results aligned closely with the findings reported by Guevara et al. [38], who found that 106 species would be classified as LC, 78 as VU, and 16 as EN in their analysis of lowland trees in the Amazon. When analyzing these findings, it was possible to observe that a large part of the species studied were from LC; this agreed with what was mentioned by Brummit et al. [89], who reported that approximately 65% of plant species globally were likely to be of Least Concern.

In our analysis, we identified one species, *Centrolobium ochroxylum*, categorized as “Critically Endangered” (CR), which was consistent with the single CR species reported by Guevara et al. [38] in the list of lowland trees for the Amazon. This species stands out as particularly threatened and requires urgent conservation attention. Additionally, we observed that 0.9% of the species were classified as “Data Deficient” (DD), indicating that there was insufficient information available to assess their threat level accurately. This highlighted the importance of conducting further research and data collection to better understand and evaluate the conservation status of these species. Globally, currently 8.1% of evaluated plant species (n = 66,468) were in the DD category due to the lack of information necessary to apply the Red List criteria. In the specific case of Ecuador, 6.7% (n = 5512) of the species were classified in this category [47]. At the EAR level, Guevara et al. [38] reported nine tree species classified as DD. Currently, there are still 390,287 plant species to be evaluated globally [90].

A relevant finding in our research was that 28.0% (60 spp.) of the analyzed timber species are not evaluated by the IUCN. Based on the global assessment of the threat status of plants provided by Brummitt et al. [89], up to 115,291 plant species are currently listed as NE. On the other hand, in relation to the EAR, this absence of evaluations was also observed by Guevara et al. [38], who pointed out that 89% of lowland tree species in the EAR are currently listed as NE. In conclusion, the IUCN Red List provides valuable insights into the conservation statuses of timber species in the Ecuadorian Amazon Region. While a significant proportion falls into the LC category, indicating relatively lower immediate risk, attention must be given to the species classified as VU, EN, and especially CR. Additionally, addressing the Data Deficient species and expanding assessments to include a broader range of plant species will contribute to more comprehensive conservation efforts in the region and globally.

### 3.4. Considerations for Current and Future Conservation of the Most Traded Timber Species in the EAR

#### 3.4.1. Promotion of Botanical Sampling and Its Digitization in an Open Access Database

Promoting botanical sampling and the digitization of herbarium specimens is crucial to enhance our understanding of the collection patterns of forest species in the EAR [38]. Herbarium specimens contain valuable taxonomic and geographic information that can significantly contribute to scientific research and the study of biodiversity. By digitizing this information and making it accessible through open-access databases like GBIF, researchers can gain access to a wealth of data for their studies, enabling a more comprehensive understanding of species distribution and ecological patterns [91]. Unfortunately, a common feature in all these biodiversity databases is the existence of biases in their structure of taxonomic and geographic information [59,60,92]. These biases can affect the representativeness of the information and limit the complete understanding of the distribution patterns of the species [59,60]. To achieve a more robust and complete understanding of forest species in the EAR, efforts must be made to improve data quality and completeness in these databases. Over more than 300 years, there has been an ongoing effort to collect taxonomic information and information on the geographic distribution of biodiversity globally [93,94]. Nevertheless, it should be noted that the mass digitization and organization of this data on accessible platforms began a little over two decades ago thanks to advances in technology that have allowed the storage and availability of this data in digital systems [94,95,96,97,98]. Despite this progress, challenges remain in terms of the quality, completeness, and standardization of the data collected [59,60]. One of the current challenges lies in the scarcity of digitized herbarium collections available for plants [96,97]. To address this issue, concerted efforts are required to prioritize the digitization of herbarium specimens and make them freely accessible for research purposes. By doing so, we can create a more comprehensive and up-to-date database of botanical information, thereby improving our ability to monitor species distribution, assess their conservation status, and inform conservation strategies.

In this context, it is imperative to intensify botanical sampling efforts to improve our understanding of distribution patterns of timber species in the EAR. Specifically, an increase in the digitization of existing botanical collections in national and local herbaria within the EAR is required. These collections contain valuable data that can significantly contribute to scientific research and conservation efforts. Making this information available through open-access platforms, both within Ecuador (BIOWEB and SISBIO) and globally (GBIF and IDIGBIO, among others), will foster collaboration and facilitate data sharing with the scientific community, leading to more comprehensive and informed analyses.

Furthermore, it is essential to promote the implementation of continuous monitoring of all forest species through the National Forest Inventory of Ecuador [99]. Therefore, it is fundamental to use methodologies that allow the identification of optimal areas in geographical and environmental terms [100]. This would allow documenting a greater number of specimens with a reduced economic investment and human effort in order to obtain a more complete and enriched database [101].

Combining intensified botanical sampling efforts, digitization of herbarium collections, and continuous monitoring will provide a more holistic and up-to-date view of the distribution patterns of timber species in the EAR. This knowledge will be invaluable for informing conservation strategies and policy-making, promoting sustainable forest management, and ensuring the long-term preservation of the region’s biodiversity.

Indeed, addressing the need for improved methodologies to enhance botanical sampling efforts and understand species distribution patterns, innovative solutions have emerged within the R programming environment. Among these, the WhereNext package [102] stands out, providing a valuable tool to analyze compositional differences between two sampling sites, considering both their environmental conditions and the geographical distance that separates them. By analyzing these features, it is possible to identify areas that minimize the average distance between previously sampled sites and those that have not yet been explored [102].

Another promising alternative is the Biosurvey package, which uses a methodology that considers the environmental conditions available in the study areas and the species’ presence records [100]. This tool allows to evaluate preselected sites and determine their climatic suitability based on species distribution models [100]. However, it is fundamental to highlight that the results obtained using species distribution models must be verified by experts in the relevant species of interest [103]. While these models can be useful tools, expert knowledge and field verification are essential to ensure the accuracy and reliability of the predictions.

To maximize effectiveness in sampling, it is essential to foster collaboration between scientists, conservationists, and decision makers, and to promote the sharing of data and knowledge on accessible and open access platforms [97,99,104,105,106]. To advance in the field of biodiversity and maximize the potential of the information available, it is necessary to continue improving the systems for collecting, storing, and accessing data [107,108]. This includes the promotion of common data standards, collaboration between institutions, and the incorporation of new technologies, such as artificial intelligence and machine learning, to facilitate the analysis and interpretation of data [95,109,110,111,112,113]. In this way, we will be able to take full advantage of the value of herbarium specimens and databases for scientific advancement and forest and biodiversity conservation.

#### 3.4.2. Current Tools to Assess the Preliminary Conservation Status of Species

The need to assess the risk of extinction of all plant species at a global and local level has been highlighted in various studies [38,114]. Nonetheless, the IUCN Red List faces significant challenges in keeping assessments up to date and reducing the proportion of data-deficient species [115]. One of our relevant findings in this study was to identify that 28% (60 spp.) of the species analyzed do not present a verified evaluation within the IUCN red list. Therefore, automated assessments based on geographic occurrence records available in digital format could be of vital importance. These automated assessments would make it possible to identify species or groups that face a higher risk of extinction, which would allow manual assessment efforts to be focused on the species that most require it [116].

The implementation of automated assessments takes advantage of the availability of digital geographic data on the distribution of species. By using algorithms and spatial analysis techniques, it is possible to identify patterns and trends in plant distribution, providing valuable information for extinction risk assessment [115,116]. At present, various tools developed for R have been created in order to automate these processes and help to be implemented on a large scale. For example, among them is the ConR package [117], which is a scientific tool that simulates an evaluation based on IUCN criteria B. This criterion implies calculating the Extent of Occurrence (EOO), which is defined as a convex hull around the areas where the presences of a species have been recorded. In addition, the package estimates the Approximate Area of Occupancy (AOO) using grid cells that cover the areas occupied by the species based on georeferenced data. In addition, the IUCNN package [118] is also available, representing an easy-to-use implementation of deep learning methods that aims to approximate extinction risk assessments of species on the IUCN Red List that currently have data-deficient or unassessed tree species [46,113].

Despite the various benefits that these technological tools provide, it is fundamental to recognize that automated assessments cannot completely replace conventional assessments, as there are additional factors that need to be considered, such as the conservation status of habitats and the specific threats facing species [119,120,121]. Therefore, it is necessary to establish an integrated approach that combines automated assessments with the experience and knowledge of conservation experts to obtain more accurate and reliable results [103,122,123].

## 4. Materials and Methods

### 4.1. Study Area

The study area centered on the Ecuadorian Amazon Region (EAR), encompassing approximately 46.85% of continental Ecuador and occupying an extensive land area of 116,687 km^2^. This region is composed of six provinces, being Sucumbíos, Orellana, Napo, Pastaza, Morona Santiago, and Zamora Chinchipe (Figure 5). It is characterized by an average annual rainfall that fluctuates between 2000 and 5000 mm and an average annual temperature of 24 °C, which makes it a region with a warm–humid climate [124].

### 4.2. Selection of Forest Species with Harvesting Records in the EAR

To identify the timber species for our study, we utilized statistics spanning from 2012 to 2022 focusing on the use and trade of timber forest species documented in the Forest Administration System (SAF) of the Ministry of the Environment, Water and Ecological Transition of Ecuador (MAATE) [125]. The SAF is based on a geographical database containing both cartographic information and relevant alphanumeric data. This tool is now fully integrated, which strengthens the management and monitoring process. In addition, the SAF database provides information specifically related to the common and scientific names of timber species traded in Ecuador, the different types of forest harvesting, and the province where the timber was harvested and the year. From this database, a filter was conducted considering the six provinces of the EAR; we considered all timber species reported as being commercially traded within the last decade. As a result, we identified a comprehensive list comprising 214 timber species that met the selection criteria for further analysis and evaluation in our research.

### 4.3. Patterns of Collection of Timber Species Used in the EAR

To address the first objective concerning the geographic collection patterns of timber species within the EAR, an extensive search for presence records was conducted across multiple global biodiversity databases. These databases included the Global Biodiversity Information Facility (GBIF), Integrated Digitized Biocollections (iDigBio), SpeciesLink (a distributed information system that integrates primary data from scientific collections), and the Botanical Information and Ecology Network (BIEN). Furthermore, to ensure a comprehensive analysis, we also explored Ecuador-specific databases, such as BIOWEB and Ecuador’s National Biodiversity Monitoring Program (SINMBIO). Finally, records reported in the Ecuadorian lowland Amazon tree database were also included as reported by Guevara et al. [38].

Throughout the data collection process, only georeferenced records reported in botanical documents were included since they were of crucial importance to clarify the taxonomy of plants and their main habitats [51,126]. On the other hand, this type of data currently has great potential for the conservation of threatened species [40,41,97]. To ensure the quality and reliability of the presence records, we applied a data cleaning protocol based on the recommendations made by Cobos et al. [98]. Initially, to eliminate duplicate samples of the same individual stored in different institutions, records with identical geographic coordinates were removed. Subsequently, records with less than two decimal places were excluded to enhance the spatial precision of the data. These steps were taken to ensure the accuracy and reliability of the georeferenced information.

To examine the geographic patterns of collections, we performed a spatial intersection between the presence records and a vector layer representing the administrative limits of the EAR. To conduct this process, the EAR vector file was obtained from the geographic data set available in DIVAGIS [127]. For the spatial intersection process, we utilized the intersect function of the terra package, specifically developed for R, which allowed us to handle vector geographic data efficiently.

### 4.4. Protection Coverage within the Current Conservation Initiatives in Ecuador

To address the second objective of determining the protection coverage for all the analyzed timber species, considering the four conservation initiatives in force in Ecuador, we acquired the vector layers corresponding to these initiatives from the special data infrastructure of the MAATE [10]. Table 2 provides an overview of the characteristics of the data used in this analysis. It is worth noting that we utilized the most recent available version for all the datasets, ensuring that the information used in our study was up-to-date and relevant.

As the next step, similar to the previous process, we conducted a spatial intersection between presence data of the studied timber species and the four current conservation initiatives. This intersection allowed us to determine the number of species and the number of corresponding observations found within each conservation initiative. By integrating the spatial information from these vector layers with the presence records of the timber species, we aimed to gain insights into the extent to which each species is covered and protected within the various conservation initiatives implemented in Ecuador.

In addition, we carried out the same counting procedure for the areas of the AER that are not currently covered by any of the initiatives. This analysis aimed to identify the percentage of timber species that could potentially be threatened due to the lack of protection in those specific areas.

### 4.5. IUCN Conservation Categories for the Analyzed Timber Species

For the fulfillment of the third research objective, we focused on determining the IUCN conservation categories for the analyzed timber species. To do this, we utilized the scientific names of the species and employed the “iucn_summary” function from the taxize package [129]. By executing this function, we conducted a query to obtain the most up-to-date IUCN conservation status for each species.

By combining the results of the IUCN conservation categories with the findings from the spatial intersections and protection coverage analyses, we aimed to gain a comprehensive understanding of the status of the timber species in terms of conservation efforts and potential threats. This holistic approach would support the development of effective conservation measures for these valuable forest resources within the Ecuadorian Amazon Region.

## 5. Conclusions

In conclusion, to promote the conservation of forest biodiversity in the Ecuadorian Amazon Region (EAR), it is crucial to undertake new botanical samplings in less-explored areas and digitize existing collections in national herbaria. These actions are essential to promote the conservation of species and provide valuable data that support decision-making in the management of forest resources. The availability of accurate and updated information will facilitate the identification and protection of species at risk.

Automated assessments of the conservation status of forest species are useful tools, especially for those species that have insufficient data and have not yet been assessed. These assessments contribute to prioritizing conservation efforts and optimizing the allocation of resources to the most vulnerable species. By using automated methods, the assessment process is streamlined, and an overview of the conservation status of various species in the EAR is obtained.

Species distribution models are highly recommended to identify suitable areas for forest restoration initiatives in the EAR, especially in the global context of initiatives such as United Nations declaration of the decade on ecosystem restoration. These models evaluate geographic locations based on relevant climatic variables, facilitating the selection of priority sites for restoration and appropriate species that can thrive under changing environmental conditions.

The effective communication of the results of the conservation assessments of forests and their biodiversities aimed at academia and those responsible for decision-making on the protection of biodiversity, which must be coordinated with existing conservation efforts in Ecuador, such as the SNAP, the BVP, the PFE, and the SB. All this makes the importance of strengthening scientific research, fostering inter-institutional collaboration and promoting the active participation of society in the conservation and management of forest biodiversity in the EAR a priority.

## Figures and Tables

**Figure 1 plants-12-03327-f001:**
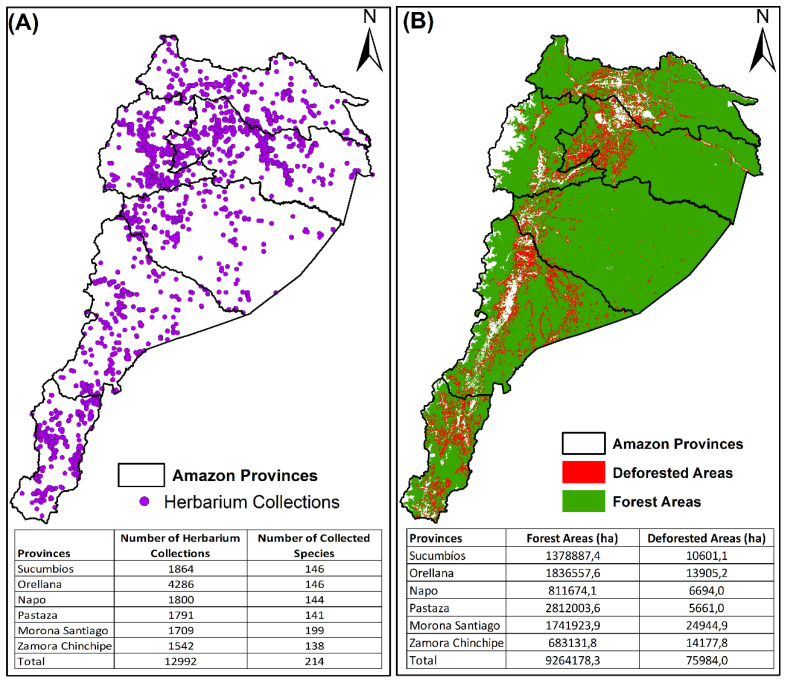
Collection patterns of species of high commercial interest from EAR. (**A**) Number of botanical collections deposited in herbarium. (**B**) Map of deforestation and native forest cover in the EAR.

**Figure 2 plants-12-03327-f002:**
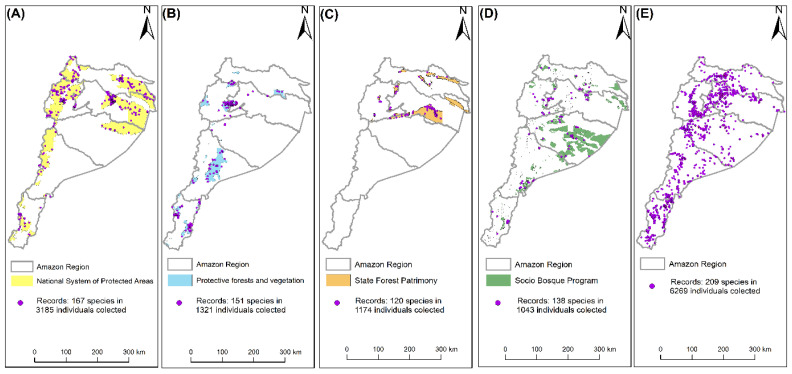
Patterns of collection of timber species of high commercial value from the Ecuadorian Amazon: (**A**) National System of Protected Areas; (**B**) Protective Forests and Vegetation; (**C**) State Forest Patrimony; (**D**) Partner Forest Program; and (**E**) Without Management Category.

**Figure 3 plants-12-03327-f003:**
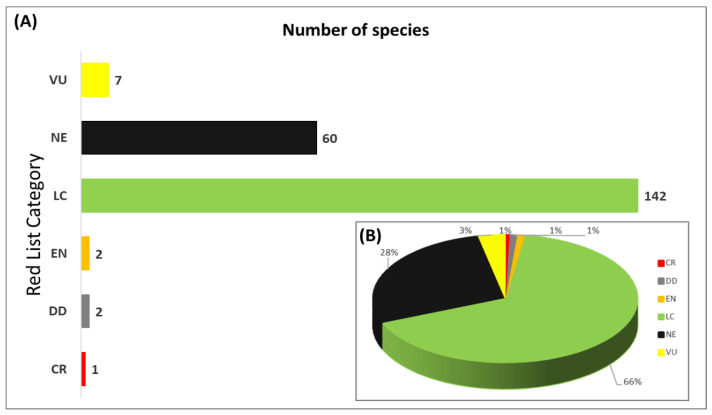
(**A**) Number of species; and (**B**) Percentage for each of the IUCN conservation categories. Categories: CR = Critically Endangered, EN = Endangered, VU = Vulnerable, NT = Near Threatened, LC = Least Concern, DD = Data Deficient, and NE = Not Evaluated.

**Figure 4 plants-12-03327-f004:**
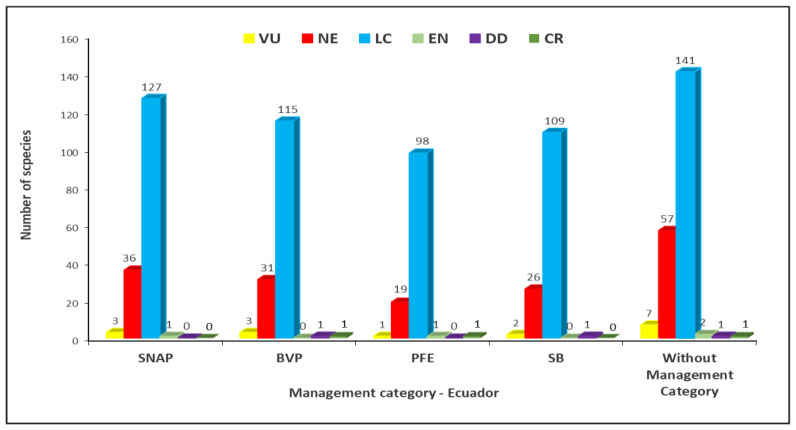
Number of species within the current management categories for Ecuador and IUCN conservation categories. Categories: CR = Critically Endangered, EN = Endangered, VU = Vulnerable, LC = Least Concern, DD = Data Deficient, and NE = Not Evaluated.

**Figure 5 plants-12-03327-f005:**
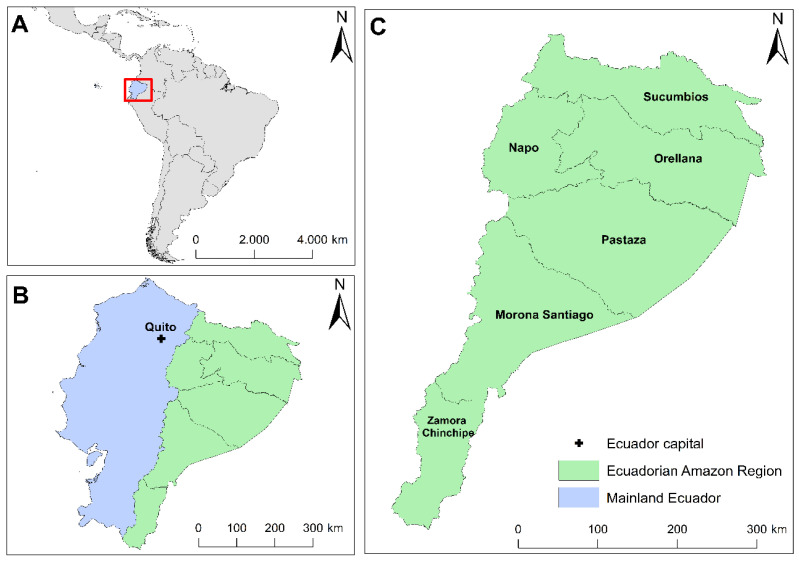
Study area: (**A**) Geographic location of Ecuador; (**B**) Ecuadorian Amazon Region (EAR); and (**C**) Provinces belonging to the EAR.

**Table 1 plants-12-03327-t001:** Number of individuals and species in the provinces of the Ecuadorian Amazon.

Province	Individuals	%	Species
Sucumbíos	1864	14.3	146
Orellana	4286	33.0	146
Napo	1800	13.9	144
Pastaza	1791	13.8	141
Morona Santiago	1709	13.2	199
Zamora Chinchipe	1542	11.9	138
Total	12,992	100	214

**Table 2 plants-12-03327-t002:** Geographic information in vector format of current conservation initiatives in Ecuador [128].

Data	Format
Protected areas	Vector (Polygons)
Forests and protective vegetation	Vector (Polygons)
State Forest Heritage	Vector (Polygons)
Socio Bosque Program	Vector (Polygons)

## Data Availability

Not applicable.

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
