# Peer review of "Botanical Collection Patterns and Conservation Categories of the Most Traded Timber Species from the Ecuadorian Amazon: The Role of Protected Areas"

_plants, 2023, doi:10.3390/plants12183327_

Round 1

Reviewer 1 Report

Authors, please address the following.

from the abstract, I think 22% is good enough.

The following article is no more than a technical report on the data and does not carry an in-depth study. I recommend submitting to MDPI. Data | An Open Access Journal from MDPI  

Author Response

Dear reviewer:
On behalf of all the authors of the article entitled: Patterns of botanical collection and conservation categories of the most traded timber species of the Ecuadorian Amazon: the role of protected areas; I thank you for your kind comments and suggestions, as they have allowed us to improve the scientific quality of the manuscript. The following is a detailed response to each of their comments.

Reviewer 2 Report

Authors may note the following for minor corrections:

1.line 612: Please write United Nations declaration of the decade on ecosystem restoration (instead of decade of restoration).

2.It would be appreciated if authors could provide the list of 214 economically important timber yielding tree species mentioned in line 132 and 514 as supplementary file?

3.Authors in the present study made an attempt regarding collection patterns of timber trees in the Amazon lowlands of Ecuador and also to evaluate the conservation status of these selected trees (214 species) according to IUCN categories.Objective 1 in the list  (line 125-126) is not clearly defined and seems irrelevant.

4.Methodology: Table 1 (column 2): Authors are suggested to provide the method for determining the number of individuals of trees in each province.   5.Line 145:Considering that the recent study revealed that 72 % of threatened species exist outside the protection boundaries , authors are requested to discuss the limitations of this study in discussion/conclusion section.This is pertinent considering that the communities have a great role to play in the conservation of  such trees present outside forests which may help in improvement of their livelihood, improve carbon sequestration through their considerable biomass and other ecosystem services.Authors may like to refer to the following publication:   

Thomas et al (2021): Trees outside forests are an underestimated resource in a country with low forest cover. Scientific Reports 11, 7919.  

6.Line 183 (Appendix I) : Not included for review.

Author Response

(The authors gave the same response as above.)

Reviewer 3 Report

This study relates timber trade records with protected areas across the Ecuadorian Amazon. The study is interesting and will be a valuable contribution to the literature. I have a few concerns with the manuscript. A few items need to be clarified. The methods, results, and discussion need to be streamlined. You have results in the methods and discussion in the results. After reading the manuscript, it is unclear how you expect timber patterns to be related to protected areas. If they are protected, wouldn't there be no harvest and thus underrepresented? A bit of discussion on this topic is needed.   Specific comments are below.

I wish you the best of luck in revising your manuscript.  

56. What is the 26%? Percent endemic?

57. …reptiles[8], and birds (citation).

59. Habitat is not the correct term. "Habitat types" would be okay, or ecoregion is fine.

80. What are these formal and informal activities? Carbon sequestration, as you mention in the following paragraph.

81-89. This paragraph covers multiple topics, none of which go into great depth. I suggest expanding and splitting it into multiple topical paragraphs.

97. Delete duplicate "increase."

103. Delete "accordingly."

108-115. Please break this sentence into two sentences.

120-121. How is this? Please explain how knowing this informs their threat status.

138. How do you know its limited data availability? Maybe they are rare. These are not the same thing.

144. Here and throughout, instead of leading with a Table or figure, write something compelling about the data and then cite parenthetically at the end of the sentence.

170-171. Here and elsewhere, you can say this type of thing in the discussion. "…it is crucial to highlight… However, on the results, it should not be leading and stick to the facts. Almost half (48.3%, 209 species) are outside the established protected categories. There are many instances of this in the results. I suggest deleting all of these and incorporating them into the discussion.

220-494. The discussion seems way too long and cumbersome. I suggest reducing by 1/3 to ¼ and focusing on the relevant components related to your data.

509-509. More detail is needed here. What are the criteria for trees to be included in the SAF?

537. Rework this to get rid of the result in your methods. Also, state this number in your results.

546-562. I'm unsure what this is, but it's not a method and does not belong here. It could be a study area or could be a result.

English quality is satisfactory. 

Author Response

(The authors gave the same response as above.)

Round 2

Reviewer 1 Report

Authors have improved the article and now I recommend the publication.